# UniNet: A Mixed Reality Driving Simulator

David F. Arppe*
Ontario Tech University

Loutfouz Zaman†
Ontario Tech University

Richard W. Pazzi‡
Ontario Tech University

Khalil El-Khatib§
Ontario Tech University

**ABSTRACT**

Driving simulators play an important role in vehicle research. However, existing virtual reality simulators do not give users a true sense of presence. *UniNet* is our driving simulator, designed to allow users to interact with and visualize simulated traffic in mixed reality. It is powered by *SUMO* and *Unity*. *UniNet*'s modular architecture allows us to investigate interdisciplinary research topics such as vehicular ad-hoc networks, human-computer interaction, and traffic management. We accomplish this by giving users the ability to observe and interact with simulated traffic in a high fidelity driving simulator. We present a user study that subjectively measures user's sense of presence in *UniNet*. Our findings suggest that our novel mixed reality system does increase this sensation.

**Keywords:** Driving simulator, mixed reality, virtual reality, passthrough, green chamber, SUMO, procedural city generation, traffic generation

**Index Terms:** Human-centered computing—Virtual reality—User studies; Computing methodologies—Interactive simulation—Simulation evaluation

## 1 INTRODUCTION

Many driving simulators have been developed, with most of them being used for driver training or research in the field of driver safety [41]. However, these simulators often present limited features in regards to traffic simulation, and user presence [10, 23, 24]. The need for high-quality Virtual Reality (VR) driving simulators with a focus on user presence is long overdue. In addition to this, a driving simulator with traffic simulation is a strong tool for Vehicular Ad-Hoc Network (VANET) research. Network simulation is commonly used in networking research, to evaluate the performance of communication protocols and algorithms. Existing simulation tools for vehicular networks focus exclusively on network simulation. A driving simulator that combines network simulation, application prototyping, and testing would be beneficial to VANET researchers. For instance, one could evaluate the performance of a communication protocol or application by using a realistic virtual environment with thousands of vehicles and interacting with them before deploying their research in the real world, which is costly, and at times, unsafe. The driving force behind our work was to create a simulator, which can bridge a gap between vehicle network research.

Virtual reality driving simulators have existed for as long as modern VR has existed [41]. Typically used for driver training, simulators have the advantage of being consistent. Simulators run real-time simulations, in which all aspects of the virtual environment are controlled. The input to a driving simulator is designed as a realistic imitation of the target vehicle, and the underlying simulator model simulates the interaction between the user and the target vehicle. Visual, auditory, and motion output are common forms of feedback that the simulator can provide to the user, to complete the simulator model. However, an issue with current VR driving simulators is the lack of user presence.

### 1.1 Contributions

In this paper, we describe *UniNet* – a driving simulator that combines realistic vehicle dynamics [39] with a high performance traffic flow simulation platform *Simulation of Urban MObility* (*SUMO*) [25]. We discuss the systems we have built within *Unity* [37], which connect external applications for a high quality driving experience. *UniNet* faithfully simulates a 2018 Ford *Focus* for use in situations where a physical vehicle is unsafe or unreasonable. The gear ratios, horsepower, top speed, acceleration, and suspension match the target vehicle completely. We build this simulator to enhance user presence in virtual environments.

We also developed an application pairing *Unity*, a commercial game engine, with *SUMO*, an industry-standard traffic simulator, to create a powerful visualisation tool for VANETs; capable of receiving real-time user interactions. The technology was used during our study, which also confirmed our hypothesis that Mixed Reality (MR) technology leads to a heightened sense of user presence. The overall result of this work also provides the foundation for more immersive MR technology, capable of a heighten sense of user presence, to be developed in future works. When listed, the main contributions of our work are the following:

1. Development of a driving simulator, which is connected in real-time to an industry standard traffic generator, and has two-way communication allowing for human interaction with the generated traffic.

2. Development of a MR technology which uses stereo passthrough vision in VR, and a green screen chamber.

3. A user study designed to evaluate the effectiveness of our MR technology by subjectively measuring user presence.

Minor work which supports our main contributions include: An algorithm which generates cities from *Open Street Maps* (*OSM*) data, a novel technique for rendering thousands of vehicles at once, and the construction of all of the hardware that supported the development of our MR technology.

## 2 BACKGROUND

### 2.1 Virtual & Mixed Reality

Modern Head Mounted Displays (HMDs) such as the Oculus *Rift* [40] bring VR to the consumer market, and the applications of VR are still being explored. The use of VR in driver training is studied by Daniel J. Cox *et al.*, who explored the effect of VR driver training with youth with autism spectrum disorder [10]. Their study explored how VR can be used to improve driving outside of a VR simulator.

MR visual displays, a subset of VR displays, are defined as merging the real and virtual worlds somewhere along the *"Reality-Virtuality Continuum"*, a scale connecting real environments with virtual environments [30]. MR is a term used to describe a VR experience on the reality-virtuality continuum, and not a specific technology which achieves this experience. Augmented Reality (AR)

---
*e-mail: david.arppe@ontariotechu.net
†e-mail: loutfouz.zaman@ontariotechu.ca
‡e-mail: richard.pazzi@ontariotechu.ca
§e-mail: khalil.el-khatib@ontariotechu.ca

technology is considered mixed reality on the reality-virtuality continuum, and can be seen used for a variety of applications, from educational displays at museums; to multiplayer smartphone games [1]. Augmented Virtuality (AV) is another form of MR, but less common than AR. Blissing *et al.* [3] explored driving behaviours in VR and a form of MR akin to AV. Their study was designed to understand how drivers' behaviours are affected by reality, VR, and MR. For their study, their MR configuration involved an Oculus *Rift DK2* HMD, with two cameras mounted onto the top, and a car. The cameras are designed to mimic the drivers' eyes, to give the user depth-perception. Their within-subjects study involved 22 participants experiencing each of their four configurations, while driving a real car. The four conditions were driving the car regularly, driving with the passthrough cameras in VR, driving with the passthrough cameras and traffic cones superimposed (MR), and full VR. The study required participants to drive a slalom course in these four configurations. The study concluded that the introduced HMD may affect driving behaviour, and that participants drove 35% slower when wearing the HMD. This particular MR configuration falls into the AR half of the Milgram *et al.* [30] reality-virtuality continuum.

## 2.2 Immersion and Presence

Often confused or substituted for one another, an important distinction exists for the terms 'Immersion' and 'Presence'. For the purpose of this literature, we use the definition of immersion as the objective level of sensor fidelity a VR system or virtual environment provides; and presence as a user's subjective psychological response to a VR system [4, 35]. It is important to measure and quantify a user's sense of presence, in order to fully understand what affects user presence in a VR environment. Insko *et al.* [19] discuss three methods for measuring user presence: Behavioural, Subjective, and Physiological.

Behavioural responses to events in VR is a form of measuring presence [12]. Freeman *et al.* [12] designed a study to measure presence using postural responses to events. Their study used first-person footage of a rally race from the hood of the rally car. The variance in posture were compared with subjective measures of presence.

Due to presence being a subjective sensation, subjective measurements of presence are the most common form of measurement [18], having even been used in Freeman's behavioural responses study [12]. Their study used the subjective responses to confirm their behavioural responses. This is because presence is an emotional sensation, and is best measured subjectively. Hence, questionnaires are the preferred method of gathering subjective measures. The Bob G. Witmer presence questionnaire is used for the purpose of measuring presence [45, 46]. A major issue with questionnaires as the primary form of measuring presence is that the user needs to take the questionnaire after the immersive experience, and the results depend on the user's memory [33]. However, the questionnaire approach to measuring presence is still preferred because questionnaires are easy to administer and analyze [19].

Physiological measurements have been used to measure a user's sense of presence. Heart Rate Monitors (HRMs) can be measured, and the change in heart rate can be affected by emotions, stress, fear, etc. [19]. Physiological measurements are very objective, but the disadvantage is that they can not be linked to the change in user presence easily [19]. Equipment required for physiological measurements can also create an unnatural environment, or suffer interference from electromagnetic fields or motion.

## 2.3 Driving Simulators

Driving simulators can be effective tools for researching due to their low cost and flexibility. Paired with a realistic traffic generator, a good driving simulator can make for an invaluable tool in traffic and VANET research, where human interaction is required. This section offers an overview of current driving simulators, VANET simulators, and traffic generators that were referenced while designing our simulator.

A driving simulator is an artificial environment, designed as a valid substitute of the actual driving experience [41]. Historically, simulators were designed for aircraft, primarily to train military pilots [21]. Unlike these early flight simulators, driving simulators today are used for much more than just driver training. They are used to assess driver safety [5], in VANETs [29] and HCI (Human-Computer Interaction) [6] research, and as an alternative to most other things that typically require a car. Most modern driving simulators are three-dimensional, with a high-quality physics simulation for the user-controlled vehicle [31]. The physics simulation is a key component of the driving simulator, and it converts user interaction with the system into signals captured by sensors through the steering wheel and pedals [21]. These signals are converted into inputs for the physics simulation, and the results from the simulation are presented back to the user in the form of computer graphics, sounds, force-feedback, and sometimes motion.

Lee *et al.* [24] built a full motion driving simulator as a 'Virtual Reality' tool, without the use of VR technology as we know it today. Their simulator recreated the visual, motion, audio and proprioceptive cues we associate with driving. At the time of its creation, the new level of immersion attained by their simulator inspired its title as a VR tool. In the past decade, driving simulators have become more accessible than ever. This is in part thanks to the video game industry, pushing driving physics and computer graphics to their full potential [31]. Our simulator is built around *Unity* [37], a high-performance game engine. The following subsections discuss some related literature which uses *Unity* as a base engine for a driving simulator. These works have inspired us to build our simulator in *Unity*.

### 2.3.1 Vehicular Ad-Hoc Networks

*Unity* is a powerful game engine on its own, but it can also be combined with *SUMO* for traffic generation, and discrete network event simulators for researching VANETs [2]. Biurrun-Quel *et al.* [2] have developed a driver-centric traffic simulator by connecting *Unity* with *SUMO*. Their process involved establishing a connection between the two programs via Traffic Control Interface As a Service (TraCIAS), allowing remote control of *SUMO*. This established connection allowed the authors to poll vehicle position, and display it in *Unity*. In our simulator we approached a few things differently, namely synchronization between *Unity* and *SUMO*, Non Player Controlled (NPC) vehicle motion, and physics simulation.

Ropelato *et al.* [34] used *Unity* as the base for a VR driving simulator. Their research into VR driver training builds on traditional driver training, using *Unity* as an engine to handle the vehicle physics calculations, render the virtual world into an HMD, and provide motion feedback with six Degrees Of Freedom (DOF). Their driving simulator took place in a virtual city generated by *CityEngine* [11], and featured AI traffic built in *Unity*.

Michaeler *et al.* [29] propose in their work a system built entirely within *Unity*. Having considered *SUMO* and discrete network event simulators, they chose to simulate Vehicle-To-Vehicle (V2V) communication within *Unity*. The justification for this was that *OMNet++* combined with *SUMO* would not run until the network calculation is finished, and was therefore unsuitable for combination with *Unity*. Their implementation relied on the Self-Organized Time Division Multiple Access (SOTDMA) protocol, and was able to simulate bad reception from distances, and building interference. Their simulation would parse road types from *OSM* [9], and generated traffic signs. This was based on both road data, and explicitly positioned signs.

An instance where *Unity* was used for visualization of data, can be seen in the works of Guan *et al.* [17]. Their software for real-time

3D visualization of distributed simulations of VANETs uses *Unity*'s powerful rendering engine, to visualize a city generated by Esri *City Engine* [11]. Their visualization software combines the affordances of a to-scale map, with the power of VANET simulations.

### 2.3.2 Mobility Models

*SUMO* [25] is an open-source traffic simulation application, along with supporting tools. *SUMO* is a microscopic traffic simulator, where vehicle 'types' defined by a file, are instantiated and given 'routes'. It performs a time-discrete simulation of traffic, for an arbitrary number of vehicles. Routes are generated externally, and assigned during run-time. Routes are paths along 'edges', which correspond in most circumstances to roads. Connections between edges can support traffic lights, and multiple edges can be assigned to a road to simulate multiple lanes.

Gonçalves *et al.* [15] explored the use of *SUMO* in conjunction with a serious game driver simulator, to test Advanced Driver Assistance Systems (ADASs). Their work relies on *SUMO* not only for its multi-agent microscopic simulation, but as a 'center-server', providing all essential information to their other systems [16]. Their initial work explored the impact of mental workload and distractions on driver performance [14].

To augment *SUMO*, supporting tools exist to generate routes, convert incompatible road networks into compatible road networks, and modify compatible networks. To perform these tasks in real-time requires a socket connection from an external application to *SUMO*. The Traffic Control Interface (TraCI) [44] API exists as a part of *SUMO*'s official release, and creates endless possibilities. For our research, we use TraCI to establish a connection between *SUMO* and *Unity*. It is not uncommon to find TraCI used to couple *SUMO* with communication simulators, such as *NS2* or *NS3* [8]. In the past decade, the TraCI protocol has been implemented in many programming languages. Our simulator makes use of a modern C# implementation of TraCI from CodingConnected [7]. *SUMO* supports multiple connections from different sources, and allows us to connect communication simulators in parallel with a visualization system.

### 2.4 UniNet Compared to Related Works

Our driving simulator was designed and implemented to enhance immersion and user presence in VR driving simulators. Existing VR driving simulators used for driver training [10, 29] lack the benefits of this technology, as it will later discussed. We show, with significant results, that a user subjectively feels more 'present' in our MR configuration of *UniNet*.

Finally, we have also designed and implemented an improved architecture for connecting *Unity* and *SUMO* where each vehicle has a two-way communication with *SUMO* from *UniNet*. Our simulator allows for user interaction and involvement with the generated traffic. Current related works, e.g., [2, 17], which connect *Unity* and *SUMO*, lack this two-way communication for human involvement in the traffic simulation.

### 3 The UniNet Driving Simulator

### 3.1 Challenges with Existing Simulators

### 3.1.1 Human Interaction

The most common tools for traffic simulation often lack the built-in functionality for user interaction in the form of a driving simulator [25], and driving simulators often do not offer microscopic and continuous road traffic simulation [21]. This is due to the fact that most traffic research can be conducted without human interaction and pure simulation. We chose to address this issue by building a simulator combining an industry-standard traffic generator, with a high fidelity driving simulator. *UniNet* is our solution to this problem. Our work is capable of running continuous traffic simulation,

with real-time human interaction. The established system allows for two primary forms of human interaction:

1. Human interaction in the form of a user-controlled vehicle

2. Human interaction from outside of the traffic simulation, in the form of commands sent to the simulator

Each form of human interaction can have significant impact on the resulting traffic simulations, and enable new forms of VANET and Human-Computer Interaction (HCI) research.

### 3.1.2 Procedural City Generation

Tools for generating cities such as Esri *CityEngine* can be powerful when used for visualizing traffic flow in a 3D world, if the correct software is used to combine it with a traffic generator [11]. We designed and implemented the functionality of tools such as Esri *CityEngine* into *UniNet*, to generate cities procedurally. This type of procedural design simplifies the process of importing real-world data for research. *UniNet* is designed to generate textured buildings and roads from *OSM* data, and use real satellite imagery from *MapBox* for terrain textures. Real locations can be used to study traffic congestion and vehicle networks, when used with supported tools such as *SUMO*. Figure 1 demonstrates the procedural generation of Manhattan, the most densely populated and detailed borough of New York City. The real world data was downloaded from a community powered database.

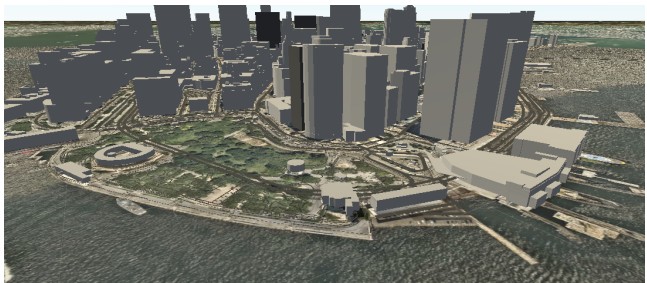

Figure 1: Procedural generation of Manhattan in *UniNet*. This image was taken after each building was blocked out, but before textures and details were added to the scene. Depending on the scale and detail of the scene, this process can take anywhere from 30 seconds to 10 minutes.

### 3.1.3 User Immersion

VR headsets encourage a new level of immersion, not often found with monitors or projectors. *UniNet*'s VR technology was designed to give the user a sense of presence in VR, or at least augment the immersion offered by consumer VR HMDs. The goal of *UniNet* is to replace traditional visual feedback with a personal VR experience, without introducing any compromises [22, 42].

Some high budget simulators have developed new ways of immersing the user, that are not always practical for smaller research [23]. An example of a configuration that is not feasible in most situations, is the use of an actual car as the cockpit for a driving simulator, designed to feature a familiar layout of controls (steering wheel, pedals) in order to not break immersion when operating the simulator [21]. Inside of a VR simulator, discrepancies between real-world controls and virtual controls may affect the user's immersion. Our novel solution is to use a stereoscopic passthrough camera, creating an MR system. Using this technology, we can superimpose the real world controls seen by the passthrough cameras onto the virtual car's dashboard.

*UniNet* also provides basic audio feedback from the user-controller vehicle, in the form of engine sounds. The sounds are controlled by the revolutions-per-minute of the engine, and the load factor on the engine. Ambient noise is provided to add realism to the simulated city and city traffic.

## 3.2 System Architecture

*UniNet* combines *Unity* and *SUMO* into a driving and traffic simulator, with many possible applications. Figure 16 offers a visual insight into how our simulator is designed. In its current form, it can simulate and render hundreds of vehicles, with user input from a physical driving simulator controlling a virtual car. At the beginning of the simulation, the user is given the option to procedurally generate a city, using real world locations as an input. The results are a full-scale copy of a real world city, that the user can drive in with virtual traffic. The traffic is generated by *Unity*, and sent to *SUMO* during the initialization phase. Each vehicle is updated by *SUMO* at a fixed time-step interval, and rendered by *Unity* to the VR headset.

## 3.3 Vehicle Physics

Our initial simulator was designed and built around *Randomation Vehicle Physics* [20], an open source vehicle physics library. The appeal was its ease of integration into the *Unity* project. However, we later swapped to *Vehicle Physics Pro* (*VPP*) [39] in favor of realism[1]. It is described as *"an advanced vehicle simulation kit for Unity 3D, providing fully realistic and accurate vehicle physics"* [39]. The integration of this physics library into our project was seamless, and we were able to focus on the technology for connecting *SUMO* and *Unity*.

The vehicle physics library was only used for the user-driven vehicle, and was not used for the visualization of traffic agents due to the complexity of the physics calculations. For this situation, we propose a *follow* technique with dead-reckoning. Each traffic agent updates their position to try and match the position and orientation of the cars simulated by *SUMO*. Due to the discrepancy in update rates, we use dead-reckoning to smooth this motion out. The *follow* algorithm follows a realistic steering model (Ackermann steering geometry) [39] to move, making for very convincing 3D movement.

## 3.4 Traffic Generation

*SUMO* [25] is an open source traffic simulator. It is capable of simulating thousands of agents traversing through a road network. It was our first choice for traffic simulation. The integration process was straightforward. For the pre-built city [36], we wrote a script to export the city map into a crude `*.net.xml` file, and used *NETEDIT* to clean it up [25]. *NETEDIT* was used to modify the direction of lanes, add traffic lights, and export the final `*.net.xml` file in the correct format for use in *SUMO*. We matched one-way streets and traffic lights with their visual counterparts in *Unity*.

*SUMO* is typically run from the console, but it could be run with the *SUMO* GUI (Graphical User Interface) option as well. We initialized *SUMO* so as to not simulate vehicle movement, unless instructed by an external process. We also set the duration of each simulated step to be 20 ms. Vehicles are added and rerouted via TraCI [44]. So it is after doing these steps that we consider *SUMO* to be fully configured. We designed *UniNet* to be the external process which commands *SUMO*.

Using an implementation of TraCI [44] in C# [7], we established a connection between *Unity* and *SUMO*. TraCI is used to populate the streets with cars from inside *Unity*, and connected each car with their agent in *SUMO*. When a user drives with the traffic simulation, a custom car is created, and labeled as an off-road vehicle type. This car is handled separately, and is mapped to the car powered by *VPP* inside of *Unity*. Its position is set each simulation update to match

---

[1] We also considered *TORCS*, an open racing car simulator) [48], as an option for vehicle physics, but decided against it due to the appeal of *VPP*.

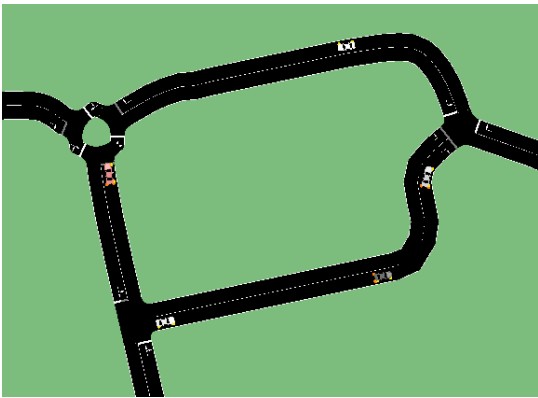

(a) City block as seen from *SUMO*

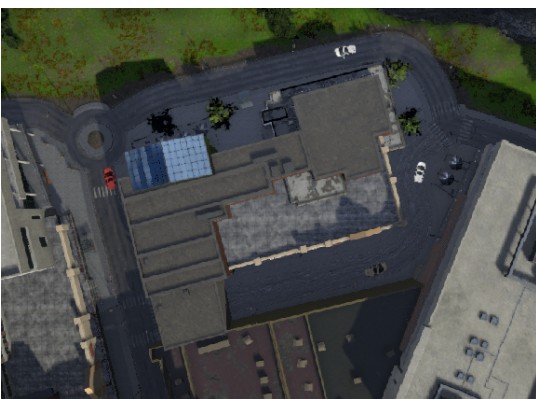

(b) City block as seen from the *UniNet* simulator

Figure 2: *Unity* and *SUMO* are seen operating together. Our simulator in *Unity* provides enhanced graphics and user interaction.

the position of the user car. In *SUMO* a vehicle can only occupy one lane at a time, so we also created a dummy car, and attached it to the rear bumper of the user controlled car. This prevents the simulated traffic agents from driving through the user's vehicle, when the rear bumper occupies a separate lane. Using *Unity*, we were able to add improved stop-sign behavior to *SUMO*. When the NPC vehicles enter a bounding box inside of *Unity*, their speed in *SUMO* is reduced to 0 for a specified amount of time. When their speed is restored, they continue as expected. Without this fix, NPC vehicles would roll through stop signs in a non-human like fashion, breaking the immersion for the driver. See Figure 2 for a side-by-side comparison of the same city block, as seen in both applications in real time.

## 3.5 City Generation

We have developed an innovative automatic city generator, which uses real world map data as input. We also support traffic simulations in fictional cities, such as *WindRidge City* [36]. The advantages to a city procedurally generated from real roads include: realistic road layouts, simple integration with map-based services, and real-time generation of 3D models based on real building footprints.

### 3.5.1 Procedural City

To generate the city from real world map data, we found that a combination of data from *OSM* [9], and procedural mesh generation techniques implemented in *Unity* was our best option. The process of creating a city starts with specifying a region, using the longitude and latitude coordinate system. From here, the simulator can download the relevant satellite imagery, land use maps, building footprints,

and roadways to create a representation of a real city. This process also works with rural and suburban areas. Algorithm 1 generates and textures 3D meshes for the roads, buildings, and terrain. All of this information is gathered from various services. *MapBox* [26] is a web service we used to download satellite imagery, heightmaps, and land-use maps. Satellite imagery is used to texture the terrain. Heightmaps are used to raise and lower the terrain, and the land-use maps are used to control placement of vegetation and bodies of water.

---

**Algorithm 1:** City Generation

**Data:** *lonMin*, *lonMax*, *latMin*, *latMax*
**Result:** Generates a city from OSM world data

1   *region ← Region(lonMin, latMin, lonMax, latMax)*;
2   *nodes ← openstreetmaps.DownloadNodes(region)*;
3   *ways ← openstreetmaps.DownloadWays(region)*;

    // Generate 3D, textured terrain
4   *texture ← mapbox.DownloadSatelliteImagery(region)*;
5   *heightmaps ← mapbox.DownloadHeighmaps(region)*;
6   *terrain ← GenerateTerrain(texture, heightmap)*;

    // Extrude buildings from footprints
7   **foreach** *Building b in ways* **do**
8      *buildingMesh ← ExtrudeBuilding(b)*;
9      *finishedBuilding ← TextureBuidling(buildingMesh)*;
10      *AddFinishedBuildingToCity(finishedBuilding)*;
11 **end**

    // Generate roads from line segments
12   **foreach** *Road r in ways* **do**
13      *roadMesh ← ExtrudeRoad(r)*;
14      *finishedRoad ← TextureRoad(roadMesh)*;
15      *AddFinishedRoadToCity(finishedRoad)*;
16 **end**

    // Add details to the city
17   *landuse ← mapbox.DownloadLanduseMap(region)*;
18   *Add3DPropsAndVegetation(terrain, landuse)*;

---

*Unity* uses a Cartesian coordinate system, and all objects exist on a flat plane on the *X* and *Z* axis. Our city generator converts geographic coordinate system *longitude/latitude* pairs, into useable Cartesian coordinate system *X/Z* pairs. The method we use to convert the coordinates is called *Mercator projection*. A drawback to the Mercator projection is that the distance from the equator will inflate distances coordinates, making distances between points inaccurate. A scalar multiplier $\theta$ is introduced and calculated based on the center of the downloaded city's bounding box. Its purpose is to scale coordinates further from the equator down, resulting in accurate distances. $\theta$ is multiplied into each of the incorrectly scaled *X/Z* pairs, and converted into a correctly scaled *X/Z* pair for use in *Unity*. We chose to scale all coordinates with the same $\theta$ value for simplicity, and as a speed optimization. We are aware that larger downloaded areas will become less accurate.

Due to floating point precision errors, we also needed a system to normalize the bounds of the city around the origin inside *Unity* (0/0/0). This was simply done by computing the *X/Z* coordinate of the center of the downloaded region, and subtracted from each future coordinate processed by the city generator.

Heightmaps downloaded from *MapBox* [26] were sampled at each coordinate, and used to generate features such as hills, riverbeds, and mountains. The sampled height was also used when generating buildings and roads, giving a third dimension to our simulator.

# 4   USER STUDY

We used a mixed factorial design user study to test if *UniNet*'s MR system improved the user's sense of presence in the virtual environment. We compared our MR system with two VR systems, and one non-VR control.

## 4.1   Participants

24 unpaid participants were recruited for the study (15 male, 9 female). Our criteria for participants was a person with VR experience, or driving experience. The participants' ages ranged from 18-57 years old ($M = 27.75$, $SD = 9.821$), with driving experience ranging from 0-41 years ($M = 9.146$, $SD = 10.417$). Of the 24 participants, 13 required corrective lenses during the virtual reality experience. 10 of our participants had used VR 1-10 times in the past, with three participants having used VR 50+ times and four participants having never experienced VR.

## 4.2   Apparatus

We used a workstation with AMD *Ryzen 5 2600x* CPU, two Nvidia *GeForce 1080 Ti* video cards, 32 GB of DDR4 RAM and 64-bit *Windows 10*. The MR simulator can be broken down into three core components: The VR headset, passthrough cameras and the green screen chamber. Vehicle input devices were used across all of the configurations. The non-VR configuration used a triple monitor setup for the output.

### 4.2.1   Virtual Reality Headset

The VR headset is an Oculus *Rift CV1*, and features a $1080 \times 1200$ Organic Light-Emitting Diode (OLED) panel for each eye, running at 90 Hz. The diagonal Field of View (FOV) of each eye is 110°, and 94° horizontally. The Oculus *Rift CV1* features constellation tracking, which is an outside-in style of tracking where infrared LEDs cover the front and sides of the headset. The accompanying constellation sensor can track the position and rotation of the Oculus HMD with sub-millimeter accuracy and near zero latency [40].

### 4.2.2   Passthrough Cameras

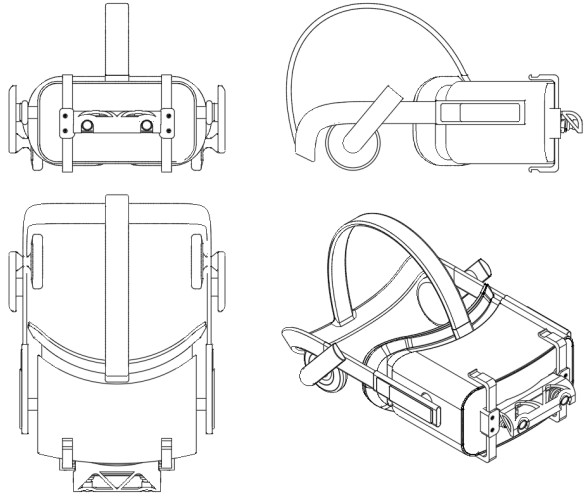

Figure 3: 3D printed mounts for the cameras allowed them to be mounted to the front face of an Oculus *Rift CV1*.

The term *passthrough virtual reality* refers to a configuration where the user can see the real world while inside a VR headset, via cameras built into or mounted on it. For our simulator we used two HD (High Definition) cameras to give the user a stereoscopic

passthrough experience. The stereoscopic camera pair are mounted to the front of an Oculus *Rift* [40], as seen in Figure 3. Properties of the camera mount are:

1. The interpupillary distance (IPD) is fixed at 60mm. This distance should closely match distance between the pupils in the users left and right eye, and 60 mm matches the human average IPD [47].

2. The downwards angle of the cameras is fixed at 5°. This is to compensate for a mismatch between the physical cameras, and the virtual cameras inside of the Oculus *Rift*, where the vertical FOV does not match the Oculus. Since our camera mount is designed specifically for a driving simulator, objects in the lower FOV (steering wheel, hands, legs) are considered more important, justifying the fixed angle of the cameras. Without this fix, the user will not see his/her arms when looking straight ahead.

3. Both cameras are parallel. Typically with stereoscopic cameras or our eyes, the stereo convergence is adjusted based on the focal point. Due to hardware limitations, we implemented a software-based solution to stereo convergence. Our left and right cameras are offset in 2D to force objects in and out of focus. This focus is then adjusted to match the stereo convergence of the virtual cameras in the headset.

The stereoscopic camera chosen is a synchronized pair of $960 \times 1280$ (960p) 60 Frames Per Second (FPS) cameras with a real time connection to *Unity*. Each camera is capable of capturing 90° FOV without distortion. The cameras are mounted strategically, in order to minimize coverage of the constellation tracking infrared LEDs on the Oculus *Rift*. The mount was 3D printed using standard black polylactic acid (PLA) filament, and conform to the front of the Oculus *Rift*. The stereoscopic camera is tilted downward 10°, in order to compensate for the lower FOV that the cameras have, compared to the Oculus *Rift*. We chose to tilt the cameras down, so that the user's legs are in their FOV while driving, because in most cases nothing is presented vertically above the user's line of sight. Figure 3 shows our 3D printed mount. The stereoscopic camera is mounted in the centre of the Oculus *Rift*, matching the height of the user's eyes. The camera has a latency of approximately 170 ms, which is compensated for inside of the game engine using a technique where the world space rotation of the headset in the virtual environment is recorded each frame. The cameras use the rotation information from the timestamp when it was captured, to choose their 3D orientation relative to the head. This allows the Oculus *Rift* and the virtual passthrough camera canvas to be synchronized. Simulator sickness was reduced by compensating for the latency of the cameras using this technique. The latency comes from the processor used on the camera's circuit board. A faster processor could encode and compress the video stream quicker, reducing the latency of the system.

### 4.2.3 Green Screen

In order to see the virtual world through the aforementioned passthrough VR system, we developed a green screen chamber, which surrounds the driving simulator completely. We use a real-time green screen algorithm run on the Graphics Processing Unit (GPU), to present the virtual world to the user in MR. For the driving simulator, this has the unique advantage that the user will see their arms and a real vehicle dashboard, while driving in a virtual city and keeping the benefits of virtual reality. Figure 4 shows a third-person view of the simulator composited with the virtual scene, and Figure 5 shows what the end-result looks like, when the video feed and virtual world are composited together.

The algorithm for the green screen is a form of chroma key compositing, to layer the captured camera feed onto the virtual

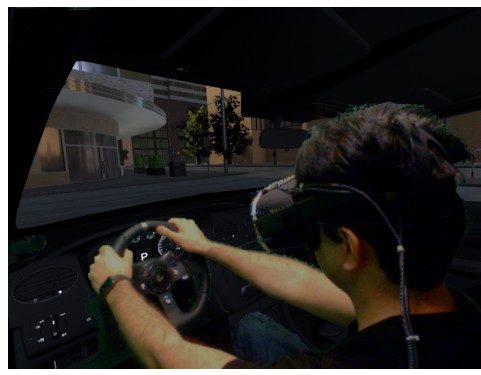

Figure 4: A third-person view of a user driving the simulator in front of a green screen, composited with the virtual vehicle.

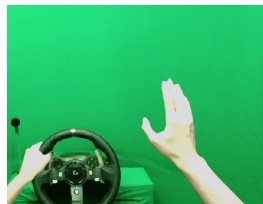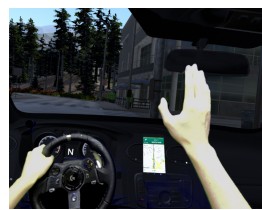

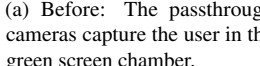

(a) Before: The passthrough cameras capture the user in the green screen chamber.

(b) After: The user is composited onto a virtual environment.

Figure 5: The simulator user can see their hands in virtual reality. This feature was added to help the user become immersed.

world. Because our algorithm is run in parallel, we chose to use difference keying instead of chroma keying. Keying is a term used when two images are composited, based on chroma ranges (color hues). Difference keying is a similar algorithm, which uses the difference between red and green pixels in the source image to composite it onto the destination image. This has the disadvantage of limiting us to using only the color green, however it is more efficient.

### 4.2.4 Vehicle Input

*UniNet* is a standard driving simulator in terms of input. Our simulator uses an off-the-shelf Logitech *G920* racing wheel, with force feedback. The clutch pedal was removed from the pedal cluster to avoid any confusion, as the vehicle we chose for the user study was an automatic transmission. Research into whether controllers affect immersion, supports our choice in a racing wheel with high quality force feedback [27, 28].

### 4.2.5 Green Screen Chamber

The green screen chamber was custom built to surround the front of the user. It surrounds ≈ 220° of the user's FOV (see Figure 6). This configuration does not cover the upper FOV of the user, however it is compensated for in code by adding a virtual green screen to the scene using the HMD rotational information. The chamber is designed to roll forward and backward on rigid casters, allowing the user easy access in and out of the simulator. LED flood lights are mounted onto the top brace of the green screen chamber. The lighting is mounted directly to the chamber, so that the orientation of the bulbs relative to the screen never changes. The bulbs are angled to light all sides of the chamber. The screen is curved to prevent shadows in corners of the fabric. This is crucial, because the

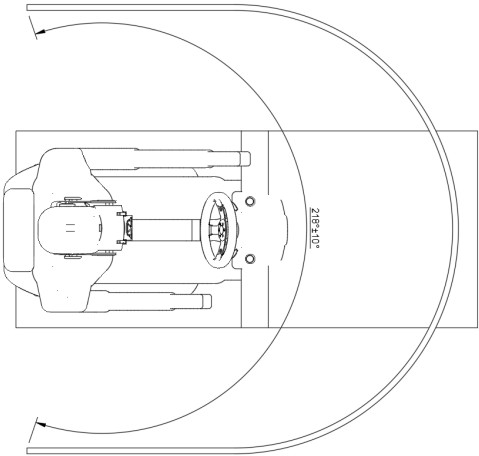

Figure 6: FOV of the user in the green screen chamber.

real-time GPU implementation of the green screen algorithm can not compensate for incorrect lighting in real time.

### 4.2.6 Triple Monitor Setup

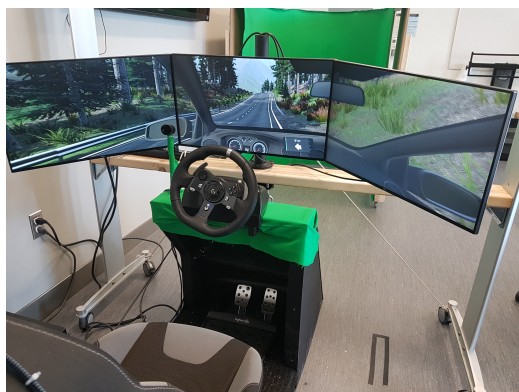

Figure 7: The triple monitor rig used for the non-VR condition.

One of the conditions in our user study used a non-VR configuration (see Figure 7). For this setup, we constructed a custom triple monitor rig, which can be wheeled in and out of position. Each monitor is $1920 \times 1080$ (1080p), with a combined resolution of $5760 \times 1080$. The rig is mounted onto a frame which can straddle the simulator. This was a requirement, in order to properly conduct our study. The experiment was counterbalanced using a $4 \times 4$ Balanced Latin square, therefore the non-VR and VR dependant conditions were constantly swapped.

## 4.3 Procedure

Participants began by completing a questionnaire about their driving experience, virtual reality experience, and demographic information. Upon completion, each user was presented a Positive and Negative Affect Schedule (PANAS) [43] questionnaire. The PANAS questionnaire is a 20 question self-report questionnaire, consisting of a 10-question positive scale, and 10-quesiton negative scale. Each item is rated on a 5-point Likert scale, and was administered to measure the positive and negative affect before the conditions began. When finished with the questionnaires, participants began the study. The participants were seated in the driver's seat of *UniNet*, and the condition was briefly explained to the participant. See below for a

description of the conditions. After each condition was completed, the participant was administered three questionnaires:

- **Bob G. Witmer PQ:** We administered this questionnaire first, as the condition was fresh in the participants mind. The questionnaire has 21 questions, taken from the Witmer presence questionnaire v3.0. The questions were chosen in order to correctly analyze four factors from the 6-factor model discussed in the original paper. The factors analyzed were Involvement (Inv), Adaptation/Immersion (AI), Consistent with Expectations (CE), and Interface Quality (IQ). The factors we excluded were Audio Fidelity, and Haptic/Visual Fidelity, because the questions were either not relevant to our research, or constant between each configuration.

- **NASA Task Load Index (NASA-TLX):** The perceived workload of each configuration was evaluated using NASA-TLX, which is a multidimensional assessment tool, and widely used to assess tasks. Total workload is divided into six subscales. Mental Demand, Physical Demand, Temporal Demand, Performance, Effort, and Frustration. A lower score on each of these subscales represents a low perceived workload for a given task.

- **PANAS:** We administered the PANAS questionnaire after each condition, and once at the beginning of the study. PANAS is used as a mood measure, in the assessment of positive and negative affect. Affectivity is a term in psychology, describing when a person is influenced by their emotions.

After all conditions and questionnaires were completed, a semi-structured interview was conducted.

### 4.3.1 Experimental Design

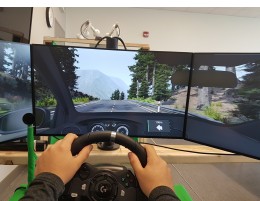
(a) The triple monitor configuration

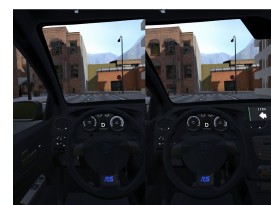
(b) The VR without hands configuration

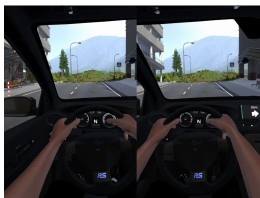
(c) The VR with fake hands configuration

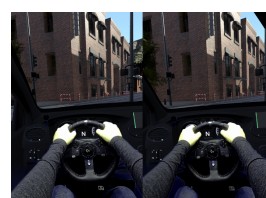
(d) The MR configuration

Figure 8: Each of the four configurations, as seen from the user's point of view.

The study was a $4 \times 4$ mixed factorial design (4 orders × 4 configurations). Order was the between-subject independent variable and was counterbalanced using a balanced $4 \times 4$ Balanced Latin square. The within-subject independent variable was Configuration. Four configurations (Figure 8) were tested as follows:

1. **NoVR**: A triple monitor non-VR control configuration, where the user is seated in front of three HD gaming monitors;

2. **VRNoHands**: A VR configuration, where the user sees the interior of the vehicle with no virtual avatar;

3. **VRHands**: A VR configuration, where the user sees a virtual avatar in place of themselves, interacting with the vehicle;

4. **MR**: A MR configuration, where the user was seated in a green-screen chamber with our passthrough VR system.

The dependent variables were Presence Questionnaire score, NASA-TLX score, and PANAS score.

### 4.3.2 City Model in the User Study

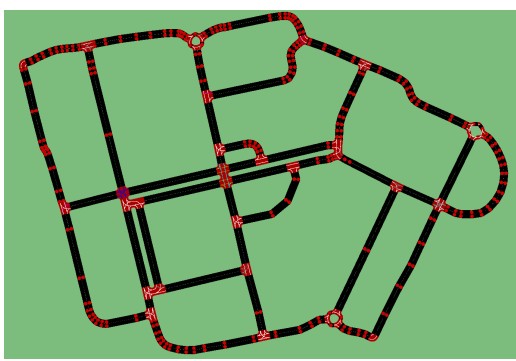

Figure 9: *WindRidge City* as seen in *NETEDIT*, showing lane directions and junctions for the simulation in *SUMO*.

For the user study, we used *WindRidge City*. This city was designed by game developers and researchers for autonomous simulation [36]. One of the advantages to using this city is its size. It contains all of the important features of a city in a relatively small footprint. In order to use this city with *SUMO*, we created a tool to map the roads inside of the *Unity* editor. This map is then exported as a `*.net.xml` file, and imported into *NETEDIT* as seen in Figure 9. It is then cleaned up, and used with *SUMO*. As a final step in preparing the city, we also swapped road signs to match the local road signs.

### 4.3.3 Conditions

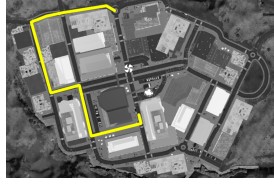

(a) The triple monitor route (510 m)

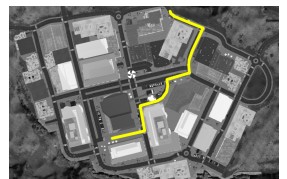

(b) The VR without hands route (430 m)

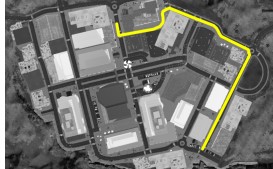

(c) The VR with fake hands route (490 m)

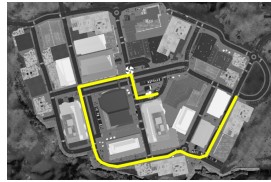

(d) The MR route (520 m)

Figure 10: Each of the routes that participants followed during the corresponding immersion configuration.

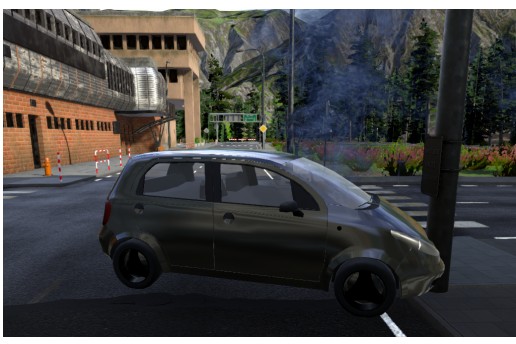

Figure 11: The spawned visceral reaction event in the Mixed Reality condition.

We designed one condition for participants to complete for each configuration: four in total. Each condition was a similar scenario in *UniNet* where the user was presented with auditory and visual navigation cues from a virtual Global Positioning System (GPS) inside of the virtual car. The GPS was mounted to the dashboard. Each condition had a unique route, and each of the four routes given by the GPS took approximately one minute. The lengths of each condition can be found in Figure 10. The conditions were completed inside of a virtual Ford *Focus*. Each aspect of the simulated car has been recreated to match its physical counterpart. Throughout the duration of each condition, the user would encounter artificial traffic. The interactions were two-way, and the user influenced traffic congestion as well as navigating through any traffic. Near the end of each condition's route, an event was spawned to generate excitement from the participant, which we believed could have an effect on the participant's sensation of presence. The events for the MR route and the triple monitor configurations, were car crashes. An NPC car would crash directly in front of the user. E.g., in MR condition, a car crashes into a fire hydrant in front of the path that the participant takes (See Figure 11). For the remaining two routes, the event was a jump-scare. An NPC car would leave a street-side parking spot as the participant was passing the parked vehicle. Both types of events instigated a reaction, either in the form of swerving or braking. The events were designed to mimic traffic collisions, to encourage a more visceral reaction when the user was more immersed. We used different events across different conditions to maintain the element of surprise. None of the conditions presented were timed, and users were allowed to take as much time as needed to finish the conditions. Given an average speed of 40 km/h, each condition takes approximately 1 minute. However we noticed that users drove faster than this limit.

### 4.4 Results

#### 4.4.1 NASA-TLX

Shapiro-Wilk normality tests revealed multiple violations of normality for NASA-TLX score. As a result, a Friedman's test was carried out to compare the NASA-TLX scores for the four configurations of the setup. A significant difference was found, $\chi^2$ (3) = 13.946, $p = 0.00298$, $W = 0.19$. A Conover post-hoc test with Benjamini & Hochberg adjustment revealed a significant difference between triple monitor ($M = 31.2$, $SD = 14.2$), and MR ($M = 27.8$, $SD = 21$), $p = 0.05$; Triple monitor and VR with fake hands ($M = 29.7$, $SD = 19.6$), $p = 0.033$. Figure 12 shows the bar plots for the overall weighted NASA-TLX scores from each condition.

Galy *et al.* propose a method of analyzing the gathered NASA-TLX data, which is to analyze the individual subscales [13]. Similarly to the overall score, Shapiro-Wilk normality tests revealed multiple violations of normality for the raw NASA-TLX scores of

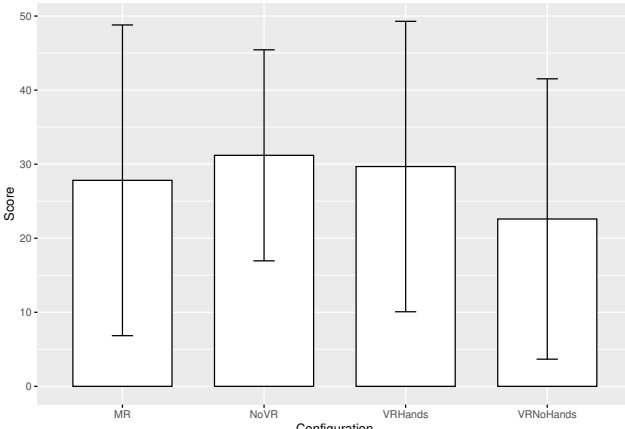

Figure 12: The bar plots for the overall weighted NASA-TLX scores. Error bars: ±1 SD.

individual subscales. As a result, just like with the overall score, Friedman tests were carried out to compare the raw NASA-TLX subscale scores for the four configurations of the setup.

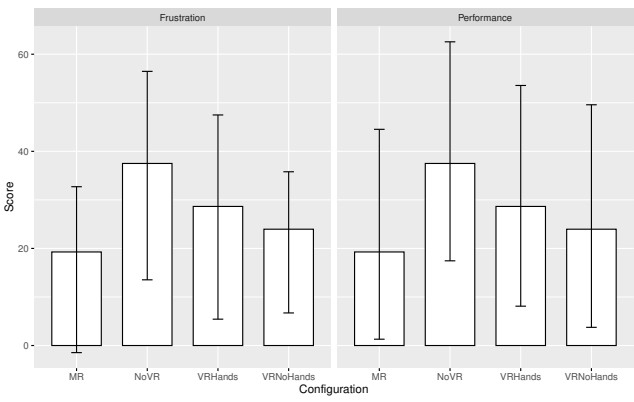

Figure 13: The bar plots for the NASA-TLX subscale scores for 'Frustration' and 'Performance'. Error bars: ±1 SD.

**Frustration:** A significant difference was found, $\chi^2(3) =$ 13.329, $p = 0.003977$, $W = 0.185$. At $\alpha = 0.5$, a Conover post-hoc test with Benjamini & Hochberg adjustment revealed a significant difference between MR ($M = 15.6$, $SD = 17.1$) and Triple Monitor ($M = 35$, $SD = 21.5$), $p = 0.0039$. At a threshold slightly above, differences were also found between MR and VR with fake hands ($M = 26.5$, $SD = 21$), $p = 0.0617$, and MR and VR without fake hands, (14.5), $p = 0.0617$. See Figure 13.

**Performance:** A significant difference was found, $\chi^2(3) =$ 8.6502, $p = 0.03432$, $W = 0.12$. A Conover post-hoc test with Benjamini & Hochberg adjustment did not reveal significant differences at $\alpha = 0.5$. Differences were found at slightly above thresholds as follows: Triple Monitor ($M = 40$, $SD = 22.6$) and VR with fake hands ($M = 30.8$, $SD = 22.7$), $p = 0.062$, MR ($M = 22.9$, $SD = 21.6$) and and triple monitor, $p = 0.059$. See Figure 13.

### 4.4.2 Bob G. Witmer PQ

Normality tests revealed no significant deviations from normality for the scores in all of the four factors: Adaptation/Immersion (AI), Consistent with Expectations (CE), Interface Quality (IQ) and Involvement (Inv). However, significant outliers were discovered for

AI and IQ. See Figure 14. As a result, we performed mixed Analysis of Variance (ANOVA) tests on CE and Inv, and Friedman test (a non-parametric alternative to repeated-measures ANOVA) on AI and IQ scores.

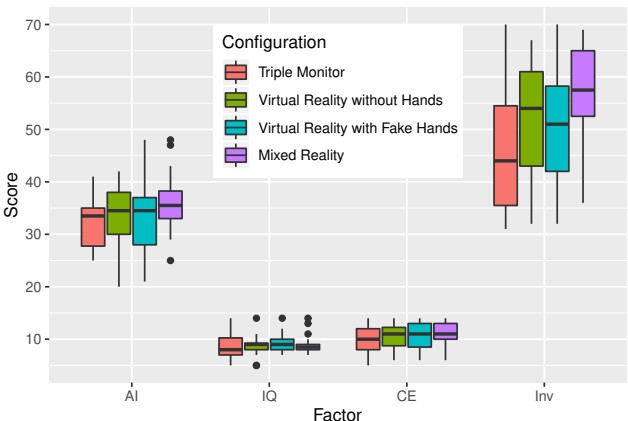

Figure 14: Scoring for four of the original six factors from the Bob G. Witmer PQ questionnaire. Due to the number of questions determining each factor, Involvement is scored from 0 to 70, Adaptation/Immersion is scored from 0 to 49, and each other factor is scaled from 0 to 14.

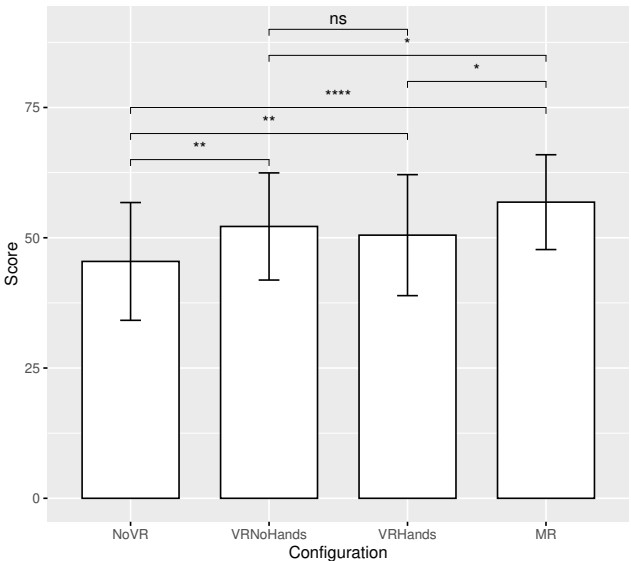

Figure 15: PQ Involvement scores for all the categories with significance levels from post-hoc pairwise t-tests. Error bars: ±1 SD. ns: p > 0.05, *: p ≤ 0.05,**: p ≤ 0.01, ****: p ≤ 0.0001.

**AI:** A significant difference was found, $\chi^2(3) = 10.92$, $p = 0.01217$, $W = 0.15$. Despite this, a Conover post-hoc test with Benjamini & Hochberg adjustment failed to reveal any significant difference between the configurations.

**IQ:** No significant differences were found between the configurations, $\chi^2(3) = 5.1659$, $p = 0.16$, $W = 0.07$.

**CE:** The main effect of configuration was not significant, $F(3, 60) = 1.93$, $p > 0.05$.

**Inv:** The main effect of order was not significant, $F(3, 20) = 0.65$, $ns$. This suggests no ordering effect was found and, as a

result, counterbalancing was successful. The main effect of configuration was significant, $F(3,60) = 10.15$, $p < 0.0001$, $\eta_G^2 = 0.14$. Post-hoc pair-wise tests with Benjamini & Hochberg adjustment were performed. Significant differences were found as follows: Triple monitor ($M = 45.5$, $SD = 11.3$) and MR ($M = 56.8$, $SD = 9.1$), $p = 0.00056$; VR with fake hands ($M = 50.5$, $SD = 11.6$) and MR, $p = 0.019$; MR and VR without fake hands, $p = 0.02687$; Triple monitor and VR with fake hands, $p = 0.009$; Triple monitor and VR without hands ($M = 52.2$, $SD = 10.3$), $p = 0.00912$. See Figure 15. The results for each individual question from the questionnaire are summarized and shown in the diverging stacked charts in Figure 17, 18 and 19.

### 4.4.3 PANAS

Shapiro-Wilk normality tests revealed multiple violations of normality for the PANAS scores in both: positive and negative affect. As a result, a non-parametric alternative to repeated-measures ANOVA was used to analyze the data. A Friedman's test was carried out to compare the affect scores for the five administrations of the PANAS questionnaires. No significant difference was found for the negative affect scores, $\chi^2$ (4) = 7.6532, $p = 0.1$. However, a significant difference was found for the positive affect scores, $\chi^2$ (4) = 12.787, $p = 0.012$, $W = 0.13$. A Conover post-hoc test with Benjamini & Hochberg adjustment revealed a significant difference between the start of the experiment ($M = 32.1$, $SD = 7.24$), and administration after the triple monitor configuration ($M = 30.2$, $SD = 9.51$), $p = 0.0074$.

Figure 20 shows that participants felt interested and excited towards the MR configuration, and at the beginning of the study. Less relevant emotions such as strength and inspiration do not see much variation between the configurations. Figure 21 shows that most negative emotions are neutral, and similar for participants among configurations. At most, only 20% of participants felt negative emotions during the study, as seen with the Distressed and Irritable questions.

### 4.4.4 Qualitative Findings

At the end of our study, we conducted a semi-structured interview. The interview questions were designed to gather user opinions about their preferred configuration, and their reasoning. Some questions were taken modified from a presence questionnaire, which we chose to not administer because it required heavy modification. We also asked questions to investigate how immersed the user felt throughout the study. For all 24 users, MR was the preferred configuration, when answering Q1 (See Table 1). For some, it was due to the novelty of the new technology, however participants explained that their inputs seemed to have more effect on the vehicle in the MR configuration. This was not the case, and the inputs to the vehicle remained the same among all configurations. This could be attributed to a heightened sense of presence and a better connection to the vehicle, compared to the VR conditions. We anticipated critical feedback regarding the low pixel density of the MR cameras and measured 150–180 ms of latency, however no users mentioned this during the study. No additional simulator sickness was experienced in MR among our participants. The sound-cancelling headphones were mentioned by 8/24 users, when asked Q4. They were described as a major contributing factor, to the reason they felt immersed, as opposed to being consciously aware of the room the study took place in. An interesting finding was that most users felt this way even with the triple monitor configuration. The lab where the study took place was very calm, and this might not have been the case if the environment was crowded with people or other distractions.

In response to Q7, all users mentioned they would drive the simulator again if given the opportunity, and 23/24 participants mentioned they would choose the MR immersion configuration if given the choice. A single participant mentioned they would only try the triple monitor configuration again if given the choice. This participant experienced mild simulator sickness, which was their reasoning. However despite the motion sickness, the MR immersion configuration was their response to Q1 due to its novelty.

The most recurring feedback we received in regards to *UniNet* was that motion feedback or haptics in the seat/pedals would improve the overall experience. This is something we plan on investigating in future works.

## 5 DISCUSSION

In our user study, we followed a mixed factorial design, to test if *UniNet*'s MR immersion system improved the user's sense of presence in the virtual environment. We are able to show that the MR configuration is more immersive, however the results are subjective, and come from the questionnaires we chose to include in our study.

Our analysis of the results supported our hypothesis, however we could not draw any conclusions from the behavioural results. Insko writes that, due to presence being a subjective sensation, subjective means of measuring presence have become the most popular [19]. Therefore our inability to corroborate the results from our questionnaires with the behavioural measurements taken, does not disprove our hypothesis.

The in-simulator portion of the study contained four conditions, designed to compare four configurations of *UniNet*. The reason we chose to compare four configurations was to compare common existing options for VR simulations, and a non-VR control with our technology. In summary, here is a brief description of why we chose these four immersion configurations.

1. **Triple Monitor**: This is the configuration most people are familiar with, and acted as a control for our study because it does not use VR technology. Instead, it relies on three monitors, with the outer monitors angled in to give the user a higher field of view.

2. **Virtual Reality without Hands**: VR without hands is another existing option featured in many VR simulators, and provides the user with an experience that is not interrupted by a virtual avatar.

3. **Virtual Reality with Fake Hands**: Providing the user with a virtual avatar is a common configuration in many VR simulators, and can help with the logic of a scene, for instance: In our configuration, the wheel of the car is turned by a virtual avatar instead of turning by itself.

4. **Mixed Reality**: This configuration is the most unique, and features our technology combined with existing VR technology. Each user is presented with a unique experience, featuring their own body as a virtual avatar in the virtual environment.

### 5.1 Bob G. Witmer PQ

The Bob G. Witmer PQ was the first questionnaire participants completed after each condition. Figure 15 shows a high level of involvement with the MR immersion configuration. The involvement questionnaire featured questions such as *"How natural did your interactions with the environment seem?"* and *"How compelling was your sense of objects moving through space?"* and *"How involved were you in the virtual environment experience?"*. These results are significant in direct comparison with all other configurations. Conversely, as a control, the triple monitor immersion configuration showed the lowest level of involvement when compared to all other configurations. Interestingly, there were no differences between the two VR configurations, which suggests the presence of avatar hand had little effect on involvement.

## 5.2 NASA-TLX

The NASA-TLX questionnaire was the second questionnaire participants completed after each condition. The purpose of the NASA-TLX questionnaire is to assess the perceived workload of a task or system. We observed significant differences between the task load index of the triple monitor immersion configuration and the VR with fake hands immersion configuration. This could be due to the fact that the 'Performance' scale on the NASA-TLX questionnaire may have been biased by the visceral reaction events that were spawned. Due to the differences in these events, the user's self-perceived performance could be viewed as unsuccessful (producing a higher score), as seen in the case of the triple monitor configuration. The VR without hands immersion configuration, may have had a simpler driving scenario, which would result in a lower score. This is due to the fact that the task load index of each condition is similar enough, that performance and frustration may be the only significant factors.

We analyzed the performance and frustration factors individually [13], and found significant differences between Triple Monitor and MR immersion configurations (at $p = 0.059$). This could be attributed to the lower FOV with the Triple Monitor immersion configuration, as we noticed worse performance among participants when turning at intersections and junctions. Users' self-perceived performance was also highest in the MR configuration. For the 'Frustration' factor, the Triple Monitor was higher than MR (at $p = 0.0617$). This could be due to the same reasons as the performance factor. Overall, performance and frustration could be signs of a heightened sense of presence in the MR configuration.

## 5.3 PANAS

The PANAS questionnaire was the final questionnaire participants filled out, before either completing the study, or beginning the next condition. It was also administered after the general information questionnaire used to gather participant information at the beginning of the study. The purpose of this questionnaire is to gauge the emotions, or mood, of the participants. The questionnaire was originally designed by taking terms with a strong connection to one dimension (positive or negative), and a weak connection with the opposite dimension. We found that the positive mood of the participants at the start of the study was significantly higher than their positive mood during the triple monitor immersion condition. The balanced design of the study means that this measured difference is likely not due to participants mood changing over the course of the study itself. The PANAS questionnaire uses a 5-point Likert scale, and we noticed high 'Interested' emotions (positive) after the start and MR immersion configuration. We also observed the highest level of 'Excitement' (positive) after the MR immersion configuration. The triple monitor configuration yielded the lowest overall 'Enthusiastic' (positive) emotion. The 'Distressed' emotion (negative) was significantly higher during the VR with fake hands condition than it was during the MR condition. This result could be due to the uncanny appearance of the virtual avatar used during the VR with fake hands immersion configuration.

Our results show a heightened sense of immersion was experienced by users in *UniNet*'s MR immersion configuration. These conclusions were drawn from the results of the Involvement factor of the Bob G. Witmer presence questionnaire, individual questions from the PANAS questionnaire, and our qualitative findings from the semi-structured interview.

## 6 CONCLUSION

As the market for VR continues to grow, the development of MR technology should grow with it. The reality-virtuality continuum is defined by the mixed-reality area between reality and virtuality, and *UniNet* was designed to fit within this range. Our work focused on the effect of user presence in a MR driving simulator, and the construction of a physical product.

The user study investigated the effect of our MR immersion configuration, on user presence. The user study hypothesized that our MR configuration would increase the user's sense of presence in the virtual environment, when compared to traditional VR and non-VR configurations. Participants were presented with four conditions to complete in *UniNet*, and each condition finished with a vehicle collision event to create a behavioural response from participants. The subjective results were significant, and in favor of our study's hypothesis.

Prior to the study, we designed and tested the hardware and software for *UniNet*. *Unity* and *SUMO* are the primary systems controlling player vehicles and NPC vehicles respectively. Our technology is built to work with the *Oculus Rift*, using commercially available stereoscopic cameras mounted to the front face of the HMD. Our software creates a passthrough VR experience with this hardware configuration. When combined with the green screen chamber constructed for *UniNet*, our technology fits on the reality-virtuality continuum as a unique mixed reality experience.

### 6.1 Lessons

The following are the steps taken to resolve issues that were encountered during the development of *UniNet*.

#### 6.1.1 Tracking and anchoring the camera stream onto the virtual world

To reduce the latency problem, we projected the camera feed in the direction that the user's head was facing at the instant the image was captured. With our configuration we had an average latency of 170 ms, and using this amount of latency as an example, we projected the camera feed relative to the virtual camera with the orientation the user's head had 170 ms prior. The result is an image that is correctly anchored to the virtual world, however is 170 ms behind.

#### 6.1.2 Lighting inconsistencies with the green screen

To improve the difference keying algorithm, our green screen was curved around the user. We chose a cloth material, and tensioned it to remove wrinkles. The green screen chamber has the ability to roll forward and backward, but to keep consistent lighting, we fixed LED flood lamps to the chamber. The lights retained their position relative to the green screen with this configuration.

#### 6.1.3 Matching the camera properties with the virtual camera properties

The FOV of the virtual cameras and *Oculus* HMD cameras are all known values, and we chose the stereoscopic camera to closely match these values. The cameras already had minimal distortion, but we still removed the distortion. Using a chessboard tracking pattern and OpenCV, we were able to remove the remaining distortion. The calibration variables received in OpenCV were used with a GPU version of the algorithm, and we prevented further Central Processing Unit (CPU) bottleneck.

### 6.2 Limitations and Future Work

We employed a single trial in each of the four experimental conditions. In hindsight, we wish we could use several trials instead and, e.g., measured reaction time to the spawned events as an objective measure of presence. Moreover, the four spawned events we used were different for each condition and not randomized. This was done to 1) facilitate the switch between conditions since the events were tied to the routes and routes were tied to the conditions, and 2) keep the duration of the study short.

Further research must be put into measuring presence quantitatively. The user study would benefit from a revisit with more focused subjective measurements, and better controlled behavioural measurements. The behavioural measurements we took could not be interpreted to their fullest potential, and similar work shows that

these types of measurements are still viable if implemented correctly [32, 38]. The behavioural results from our study did not show significant results, and our collection of behavioural data could be improved greatly. The hardware for *UniNet* could be improved with time, and simplified. The current iteration of the hardware has a limited FOV and camera resolution, which can be improved upon with better hardware.

It is also important to further research the impact of latency and camera properties on user presence in MR. Throughout our user study, users experienced camera latency of over 150 ms, with no negative side effects or additional simulator sickness. Furthermore, our green screen had a sub-par lighting configuration, and shadows caused artifacts near the bottom of the user's peripherals.

*UniNet* has the potential to be paired with VANET specific applications for networking research, which was tested but not fully explored. Future work could explore the use of *UniNet* in this academic context, and how user interaction in real time can affect V2V networks.

Future work can study methods of enhancing the green screen algorithm, via disparity mapping from the stereoscopic camera rig used for the passthrough VR. This would solve the current problem of lighting issues, as both depth mapping technology and green screen technology could create a better key for compositing the user onto the virtual environment.

Future work could also explore the use of motion feedback presented to the user, in the form of a full motion simulator. This would require a complete rebuild of *UniNet* from the ground up, with a different purpose in mind. The motion feedback was the most common feedback received from participants of the user study.

## 7 ACKNOWLEDGEMENTS

This project is partially funded by NSERC Collaborative Research and Training Experience (CREATE) and NSERC Discovery.

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

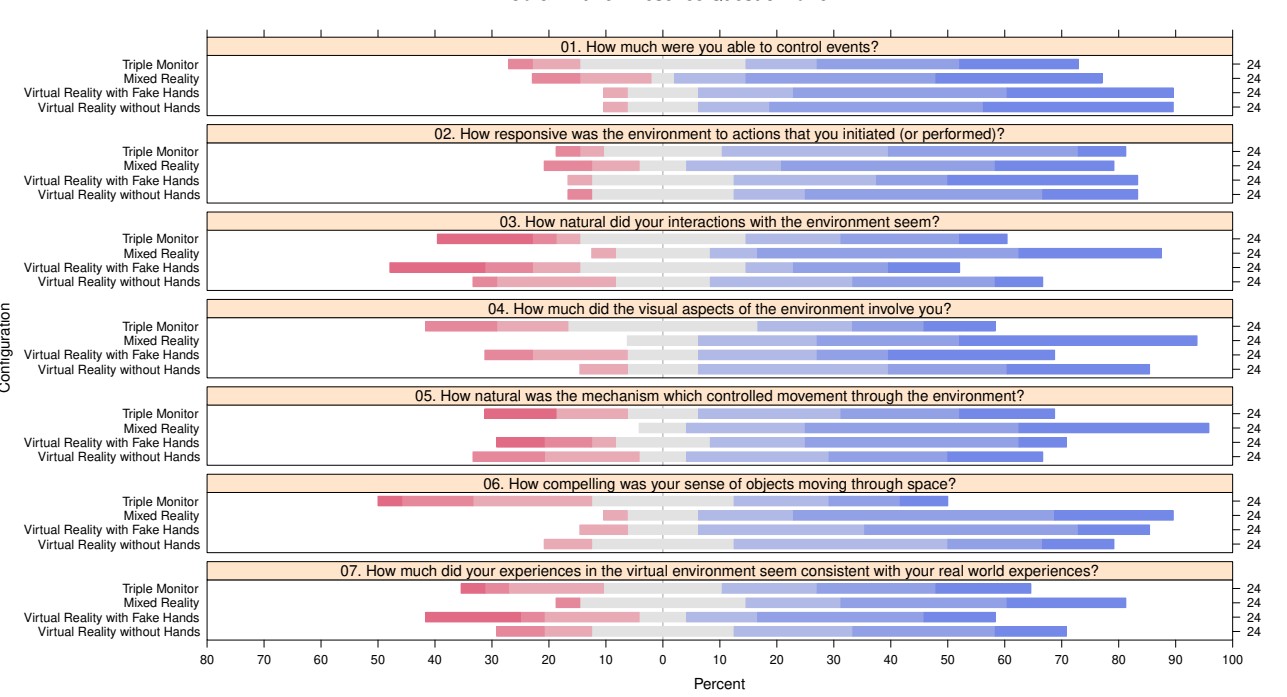

Figure 17: The Bob G. Witmer Presence Questionnaire was administered after each condition for each participant (Q.1 - Q.7).

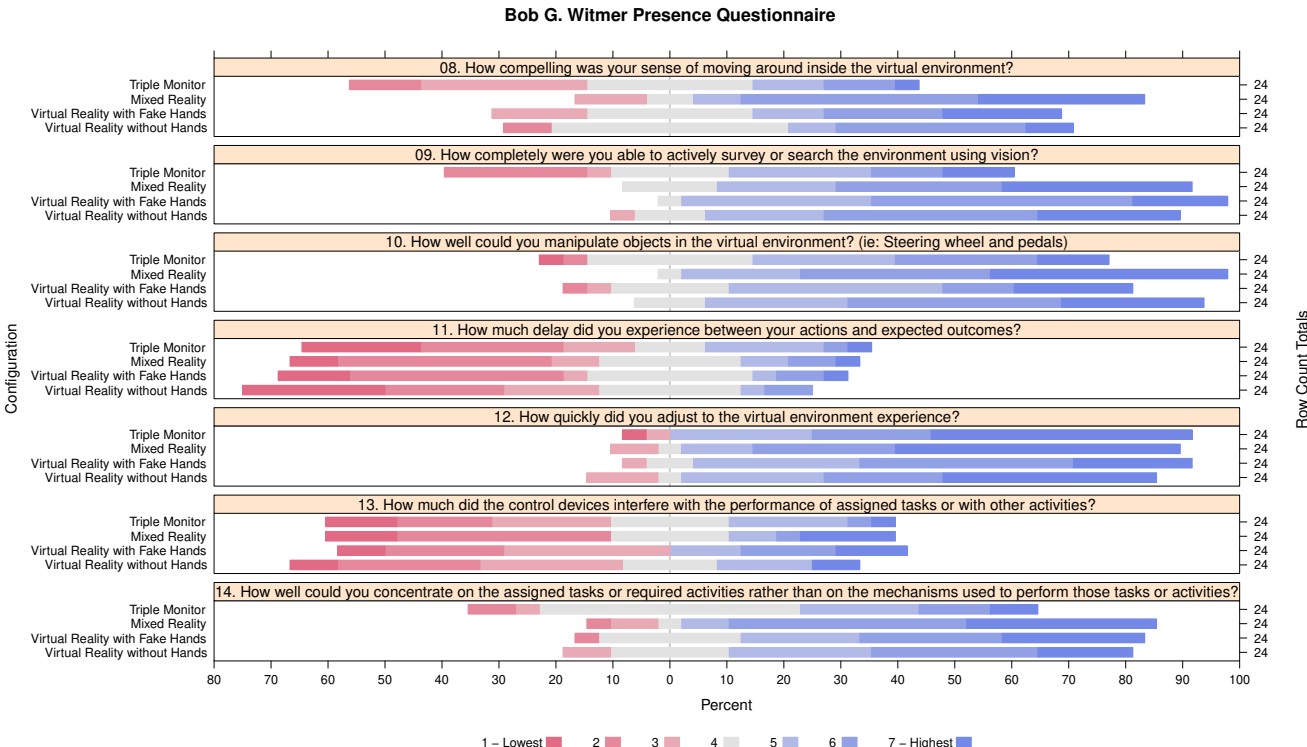

Figure 18: The Bob G. Witmer Presence Questionnaire was administered after each condition for each participant (Q.8 - Q.14).

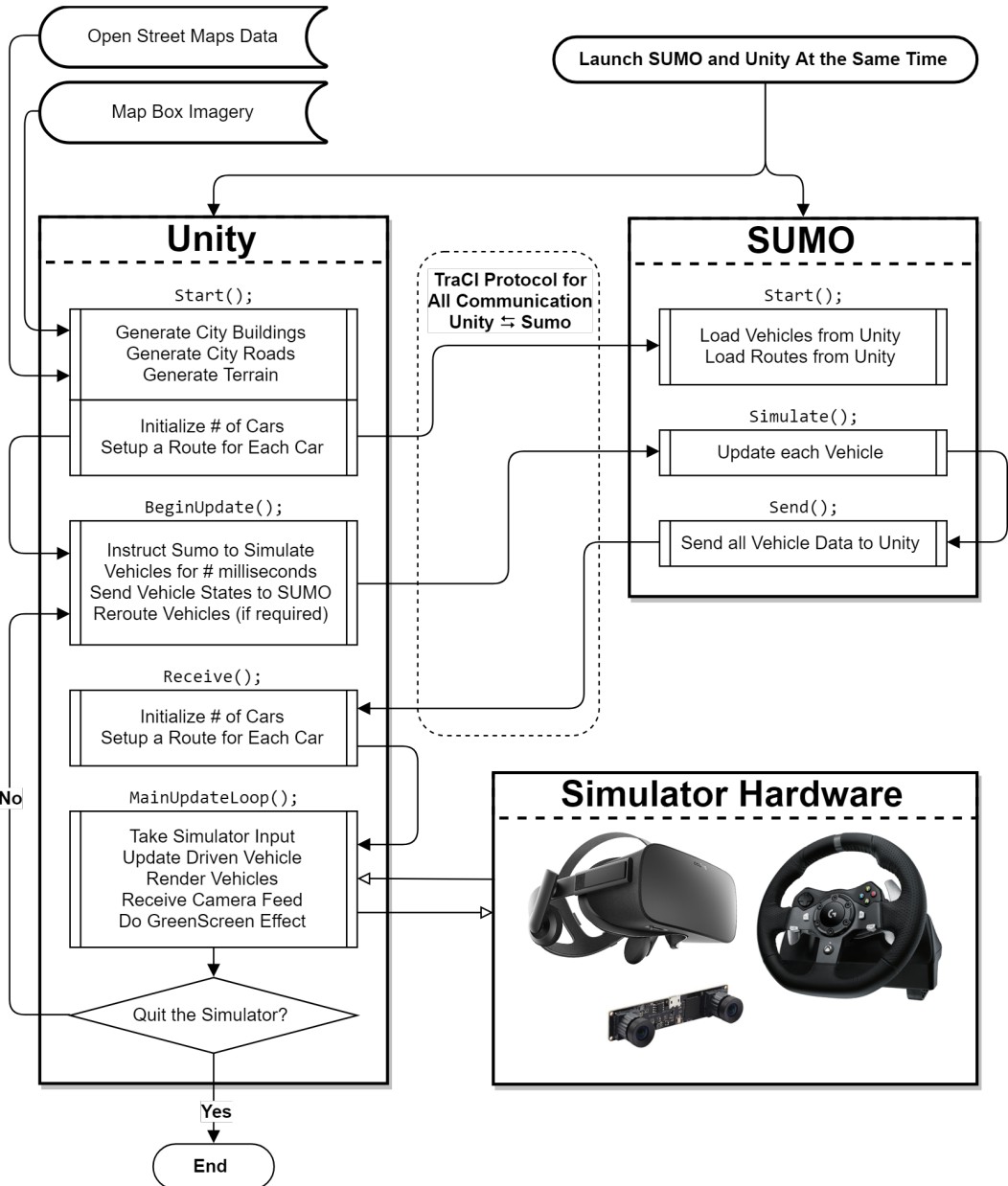

Figure 16: The *UniNet* architecture. Combining *SUMO*, *Unity*, and the supporting simulator hardware. This is a simplification of the architecture, meant to highlight the flow of data and execution of commands. The protocols interfacing *Unity* with the simulator hardware were omitted. *TraCI* is used for communication between *Unity* and *SUMO*.

**Bob G. Witmer Presence Questionnaire**

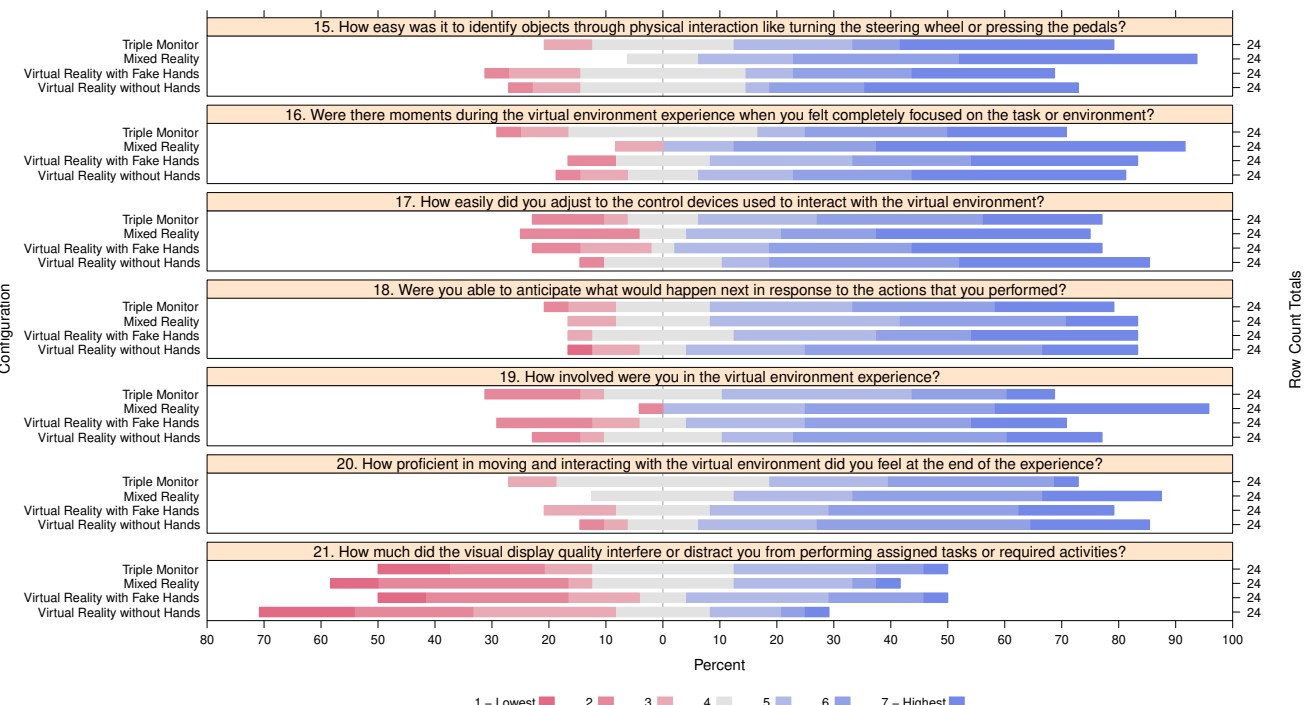

Figure 19: The Bob G. Witmer Presence Questionnaire was administered after each condition for each participant (Q.15 - Q.21).

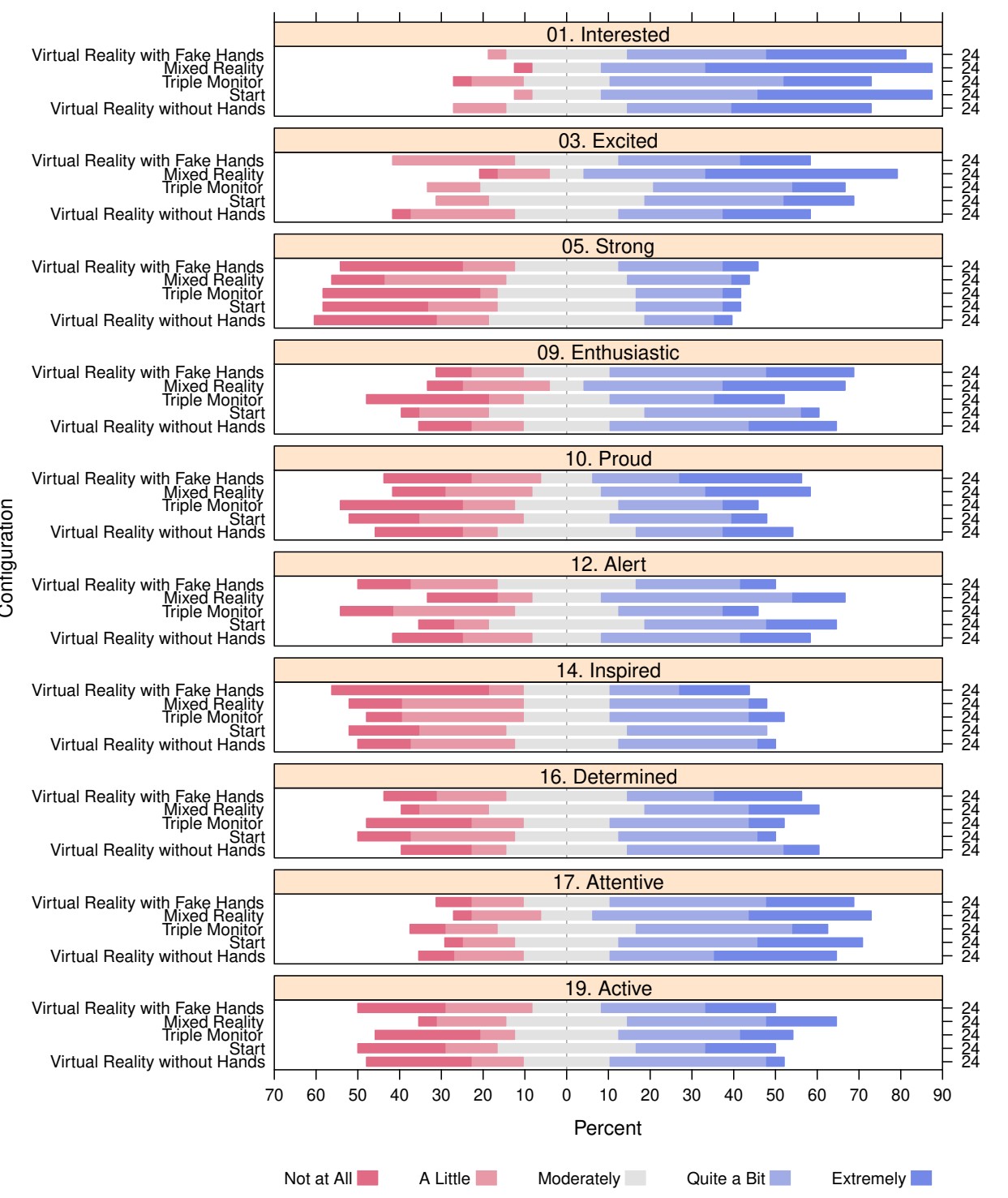

Figure 20: PANAS Positive results.

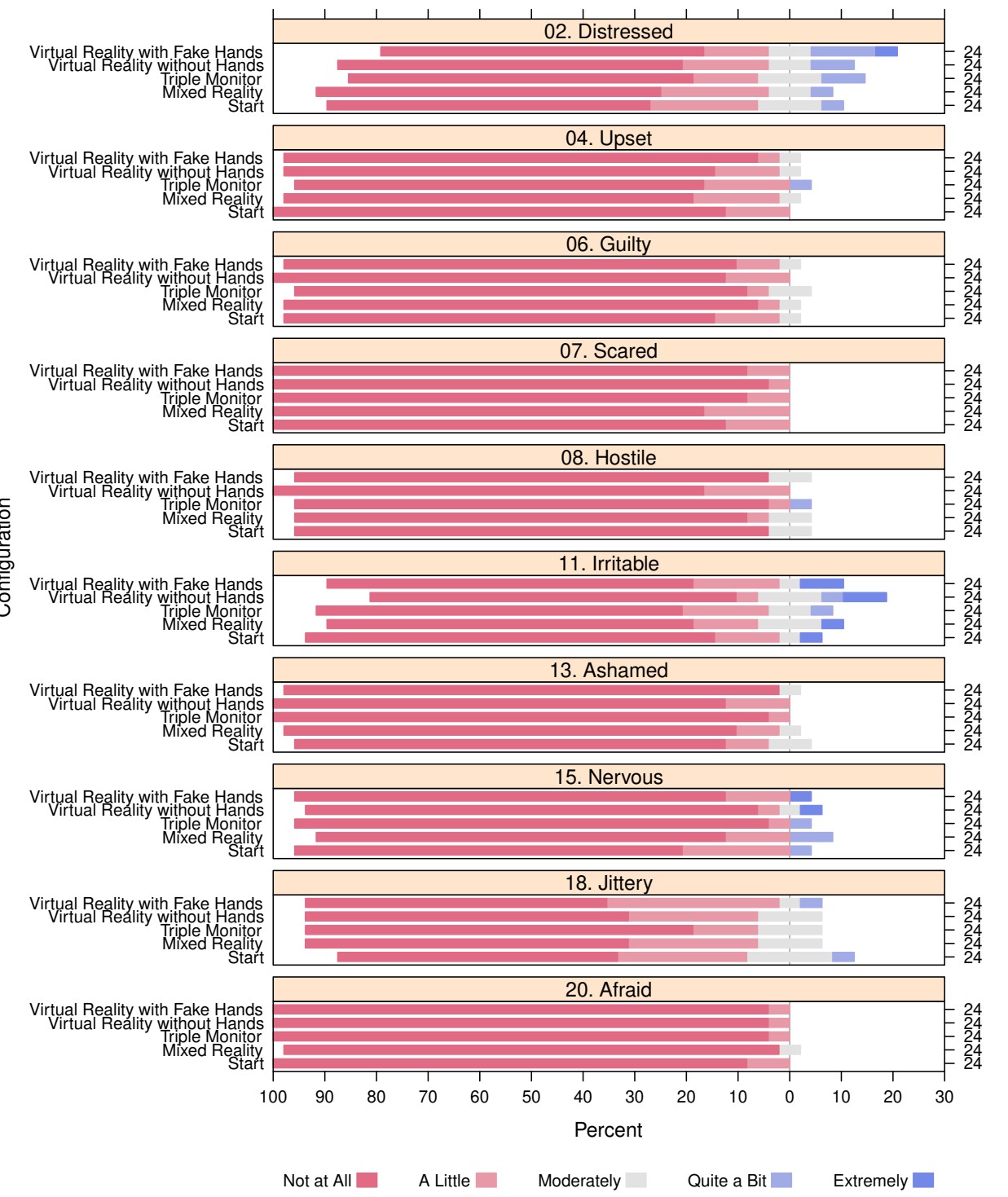

Figure 21: PANAS Negative results.

| | Question |
|---|---|
| **Q1** | Can you elaborate about which immersion configuration you liked more and why? |
| **Q2** | Can you elaborate which immersion configuration you disliked and why? |
| **Q3** | To what extent did the simulation hold your attention? |
| **Q4** | To what extent did you feel consciously aware of being in the real world whilst driving? |
| **Q5** | To what extent were you aware of yourself in the virtual environment? |
| **Q6** | To what extent did you feel that the simulation was something you were experiencing, rather than something you were just doing? |
| **Q7** | Would you like to drive the simulator again? If so, which immersion configuration? |

Table 1: The verbal script for the semi-structured interview administered after the study.