# OpenReview forum: "UniNet: A Mixed Reality Driving Simulator"
_graphicsinterface.org/Graphics_Interface/2020/Conference — GI 2020_

### Official Review · AnonReviewer2 · 2020-01-07
**The paper describes a mixed reality driving simulator that incorporates traffic generation and also claims an enhanced "presence" due to an MR system.**

**Confidence:** 3
**Rating:** 7

**Review:**

The designed system presents a serious engineering effort and addresses a genuine problem area which can be very useful. The system design is described clearly, and so is the user study and the various designs that were evaluated. However, a video would have helped here to understand the differences.

I do not have an expertise in driving simulation work and cannot judge the novelty of the MR approach for driving simulation. However, assuming it is novel, this work does present a significant contribution to the literature.

Given that the contribution of the paper is in the MR system that leads to enhanced presence for a driving simulator and in the inclusion of the traffic simulation, it is unclear if GI is the right venue for such a work. While there are elements of HCI here, the authors would benefit more from presenting this work at a venue which looks at driving simulation more carefully.

Some minor points-

1. The paper is highly redundant in its descriptions and writing and can be compressed significantly.

---

### Official Review · AnonReviewer3 · 2020-01-07
**Title?**

**Confidence:** 3
**Rating:** 6

**Review:**

This paper presents a mixed reality driving simulator setup. The objective is to enhance the sensation of presence. The description of the system is detailed and interesting. Overall this is an interesting read.

I am surprised that the introduction  says that “these modern forms of VR are relatively new". As far as I understand it, they refer to AR and MR. However authors cite Witmer's 1994 paper, which makes it not modern at all. However I agree there is a recent rise of interest in these techniques in the past few years. Overall the paper is long (14 pages + refs + appendix). It is lengthy at parts. For example it details the full history of Virtual Reality. While it is interesting, it moves the focus out of the scope of the paper in my opinion.

My few remarks are essentially about the user study. Systems are hard to evaluate overall, especially as this one is intended to be a generic tool rather than a particular application. The study about presence is a good choice. Embodiment would be another good option (Gonzalez-Franco & Peck 2018). However, the studies has only 24 participants, spread on 3 conditions in a between-subjects design, and only one trial per condition. By the way, there sees to be a vocabulary confusion in the paper. I believe that what authors name trial is actually a condition. This is is tricky because there is only one trial per condition, with only one event. This makes it difficult to make a fair statistical analysis. I suggest authors to remove everything related to reaction time because of that.
Then I wonder why different conditions had different routes and events? As far as I understand it is unnecessary because of the between-subject design, and it complicates the analysis.

In summary, There is an interesting implementation effort, and such a tool can be useful for research on interactions with cars. I am just wondering if GI is the best venue for such a work. AutoUI or VR conferences would be a better match.

Other details:

On the presentation of results, Figures 11 and 12 should be bars with 95% intervals, because of the ANOVA analysis. Figure 13 should be grouped by factor rather than condition, because we would rather like to compare conditions, not factors.

Some references have formatting issues: 8, 21, 28, 39, 51, 34 has initials instead of first names, and last name should appear before first names.

---

### Official Review · AnonReviewer1 · 2020-01-08
**This paper is a bit long with redundant paragraphs; but the contribution is enough**

**Confidence:** 3
**Rating:** 6

**Review:**

This paper proposed a complicated system for driving simulation, and apparently a great a mount of work was involved. I found it interesting to read the technical details and interview comments. The system contribution of UniNet was clearly illustrated.

My major concern was the user study. I am a bit confusing about why "MR" and "Triple Monitor" conditions are using the "car-crash" event, while "VR with hand" and "VR without hand" conditions are using the "jump-scare" event. This setting make it less convincing to compare the response time between conditions, not to mention that each condition only involved one event during the experiment.

In addition, the writing of this paper can be condensed, e.g., the material of the camera mount and working voltage/current seem to be less related to this task. Other minor comments can be found below.

(Other minor comments)
Pros:
• Great amount of details were provided to help readers understand the technical difficulties, such as the map projection distortion, VR HMD and camera synchronization, "lessons learnt" section, etc. These tips would also help other researchers & developers in this field.
• The qualitative findings from the interview & the discussion section were interesting to read.
Cons:
• The "Introduction" section can be condensed, e.g., the "Motivation" and "Purpose" parts can be merged.
• In general, the related work section was interesting to read, but many paragraphs are redundant and can be condensed. E.g., the "Mixed Reality" section, and the 3rd paragraph of the "Immersion and Presence" section.
• The structure of the system architecture can be improved -- now readers have to combine "System Architecture" on p5, "Hardware" on p7, and "Apparatus" on p8. Some paragraphs also provided duplicate information about the system, which can be better worded as well.

---

### Meta-Review · Area_Chair1 · 2020-01-09

**Recommendation:** Accept
**Confidence:** 3

**Metareview:**

There is a concern if GI is the right venue for such a contribution and whether the authors will benefit more from submitting this to a more apt venue. That said, all reviewers agree that the contribution has merit for acceptance.

---

### Decision · Program_Chairs · 2020-01-11

Accept